# Neutralization capacity of antibodies elicited through homologous or heterologous infection or vaccination against SARS-CoV-2 VOCs

Meriem Bekliz [1], Kenneth Adea[1], Pauline Vetter[2,3], Christiane S. Eberhardt[4,5,6], Krisztina Hosszu-Fellous[2,3], Diem-Lan Vu[2], Olha Puhach [1], Manel Essaidi-Laziosi[1], Sophie Waldvogel-Abramowski[7], Caroline Stephan[7], Arnaud G. L'Huillier[8], Claire-Anne Siegrist [4], Arnaud M. Didierlaurent[4], Laurent Kaiser [2,3], Benjamin Meyer [4✉] & Isabella Eckerle [1,2,3✉]

Emerging SARS-CoV-2 variants raise questions about escape from previous immunity. As the population immunity to SARS-CoV-2 has become more complex due to prior infections with different variants, vaccinations or the combination of both, understanding the antigenic relationship between variants is needed. Here, we have assessed neutralizing capacity of 120 blood specimens from convalescent individuals infected with ancestral SARS-CoV-2, Alpha, Beta, Gamma or Delta, double vaccinated individuals and patients after breakthrough infections with Delta or Omicron-BA.1. Neutralization against seven authentic SARS-CoV-2 isolates (B.1, Alpha, Beta, Gamma, Delta, Zeta and Omicron-BA.1) determined by plaque-reduction neutralization assay allowed us to map the antigenic relationship of SARS-CoV-2 variants. Highest neutralization titers were observed against the homologous variant. Antigenic cartography identified Zeta and Omicron-BA.1 as separate antigenic clusters. Substantial immune escape in vaccinated individuals was detected for Omicron-BA.1 but not Zeta. Combined infection/vaccination derived immunity results in less Omicron-BA.1 immune escape. Last, breakthrough infections with Omicron-BA.1 lead to broadly neutralizing sera.

---

[1] Department of Microbiology and Molecular Medicine, University of Geneva, Geneva, Switzerland. [2] Division of Infectious Diseases, Geneva University Hospitals, Geneva, Switzerland. [3] Geneva Centre for Emerging Viral Diseases, Geneva University Hospitals and Faculty of Medicine, Geneva, Switzerland. [4] Center for Vaccinology and Neonatal Immunology, Department of Pathology and Immunology, University of Geneva, Geneva, Switzerland. [5] Division of General Pediatrics, Department of Woman, Child and Adolescent Medicine, Faculty of Medicine, University of Geneva, Geneva, Switzerland. [6] Center for Vaccinology, Geneva University Hospitals, Geneva, Switzerland. [7] Transfusion Unit, Department of Medicine, Geneva University Hospitals, Geneva, Switzerland. [8] Pediatric Infectious Diseases Unit, Department of Women, Child and Adolescent Medicine, Geneva University Hospitals and Faculty of Medicine, Geneva, Switzerland. ✉email: Benjamin.Meyer@unige.ch; isabella.eckerle@hcuge.ch

n late 2019, Severe Acute Respiratory Syndrome Coronavirus 2 (SARS-CoV-2) emerged causing a pandemic that led to an unprecedented international health crisis of yet unknown outcome[1,2]. Shortly after its emergence, SARS-CoV-2 acquired the D614G mutation in the spike protein in February 2020, which quickly replaced all other circulating variants and spread worldwide. The evolutionary advantage of D614G is associated with enhanced binding to the human receptor and increased replication and thus presumably better transmissibility[3]. After largely uncontrolled transmission on a global scale during 2020, the emergence of the first variants of concern (VOCs) was observed[4]. The VOCs currently consist of Alpha, Beta, Gamma, Delta and most recently (late 2021) Omicron[5]. VOCs are characterized by a rapid increase in case numbers and quickly outcompeting of earlier strains in their region of emergence. Omicron has the most mutations observed so far, with the majority of them located in the spike protein[4]. Thus, the risk of Omicron escaping antibodies raised against earlier variants and vaccines is high.

In addition to VOCs, other less concerning variants were classified as variants of interest (VOI) due to aspects in their epidemiology or genetic signatures potentially leading to an altered phenotype, among them the (former) VOI Zeta that arose in parallel with the Gamma VOC in South America at a time when a surge in local cases was observed, but it has since disappeared[6–10].

Currently, few treatments are widely available for SARS-CoV-2, and are mostly reserved for specific risk groups, therefore prevention and protection through immunity generated through vaccination is still the preferred method for managing the pandemic[11]. Depending on the country, medium to high levels of population immunity have already been reached through vaccination or infection, but there are huge geographical differences when it comes to the infected proportions of the populations, variant circulation history, percentage of vaccinated individuals, vaccine doses, and type of vaccine used.

In light of that, it is particularly important to evaluate the neutralizing potential of elicited antibodies against clinical isolates of VOCs/VOIs to detect immune escape variants early and understand the impact of such variants on the further course of the pandemic.

The mRNA-based vaccine Pfizer-BioNTech BNT162b2 encodes a stabilized full-length SARS-CoV-2 spike ectodomain derived from the Wuhan-Hu-1 genetic sequence and elicits potent neutralizing antibodies[12], as does the mRNA-based vaccine Moderna mRNA-1273[13]. However, emerging SARS-CoV-2 variants include multiple substitutions and deletions in the major target of neutralizing antibodies, the spike glycoprotein, including the N-terminal (NTD) and receptor-binding domains (RBD), with the largest number of mutations (over 30) observed in the Omicron variant. This raises the question of whether neutralizing antibodies induced by early circulating strains or by current vaccines can effectively neutralize recently emerged virus variants. Currently, studies indicate that mutations that have been accumulating in the spike protein, especially in the RBD, are associated with increased affinity to the human ACE2 receptor[14], as well as with resistance to neutralization from antibodies of previously infected or vaccinated patients[15]. Mutations in the RBD pose the greatest risk for increased infectivity or immune escape[14]. Virus neutralizing antibody levels have been shown to be a correlate of protection from SARS-CoV-2 but more insights on neutralizing responses against emerging virus variants are needed[16–19].

In this study, we investigated the neutralizing potency of a panel of authentic sera or plasma from individuals vaccinated twice with either BNT162b2 or mRNA-1273, and convalescent patients that had mild coronavirus disease 2019 (COVID-19). Patients were infected at different time points during the pandemic with either an early-pandemic (pre-VOC) SARS-CoV-2 or one of the VOCs: Alpha, Beta, Gamma, Delta or Omicron-BA.1. We used authentic clinical isolates for pre-VOC SARS-CoV-2 (Pangolin lineage B.1), Alpha, Beta, Gamma, Delta, Zeta and Omicron-BA.1, which were all isolated from patient samples collected from our routine diagnostic laboratory. We assessed the neutralizing potential against homologous and heterologous variants by live virus plaque reduction neutralization tests (PRNT), widely regarded as the gold standard for the detection of SARS-CoV-2-specific neutralizing antibodies[20].

## Results

To evaluate the neutralization capacity of SARS-CoV-2 specific antibodies against SARS-CoV-2 variants (pre-VOC, Alpha, Beta, Gamma, Delta, Zeta and Omicron-BA.1), a panel of convalescent blood specimens were used from (i) individuals previously infected with pre-VOC SARS-CoV-2 ($n = 34$), (ii) individuals previously infected with VOCs Alpha ($n = 12$), Beta ($n = 8$), Gamma ($n = 10$), Delta ($n = 10$) (iii) BNT162b2 or mRNA-1273 vaccinated individuals with ($n = 6$) and without prior infection ($n = 16$) (iv) BNT162b2 or mRNA-1273 vaccinated individuals with a break-through infection with either Delta ($n = 13$) or Omicron-BA.1 ($n = 11$) (Tables 1–3).

**Neutralizing capacity from infection-derived convalescent samples.** Geometric mean PRNT$_{90}$ titers of convalescent specimens from individuals infected with pre-VOC SARS-CoV-2 were 37.3 (95%CI: 25.4–54.9) against B.1, 16.7 (95%CI: 9.7–28.8) against Alpha, 14.0 (95%CI: 8.8–22.3) against Beta, 10.3 (95%CI: 6.4–16.6) against Gamma, 12.0 (95%CI: 7.0–20.3) against Delta, 1.4 (95%CI: 0.9–2.4) against Zeta, and 0.8 (95%CI: 0.6–1.2) against Omicron-BA.1. Compared to B.1 this results in a fold-

**Table 1 Characteristic of convalescent individuals' samples.**

| Infecting virus | Number patient | Gender (M/F) | Age, mean | DPP/DPOS | Sample type | Infection period |
|---|---|---|---|---|---|---|
| pre-VOCs | 34[a] | 20/14 | 31 | 32 (25–37)** | Serum, plasma | March–June 2020 |
| Alpha | 12 | 8/4 | 51 | 27 (8–42) | Serum | December 2020–February 2021 |
| Beta | 8 | 2/6 | 42 | 50 (3–98) | Serum | January–May 2021 |
| Gamma | 10 | 4/6 | 44 | 54 (7–137) | Serum | January–April 2021 |
| Delta | 10 | 4/6 | 42 | 71 (9–118) | Serum | May–July 2021 |

*DPP* days post PCR diagnosis, *DPOS* days post symptom onset (marked with **).
[a]8/34 pre-VOC convalescent specimens here were available as anonymized left-over plasma specimens from apheresis collection of plasma (all collected in 2020), under the general informed consent of the University Hospitals of Geneva. Although specimens are anonymized, according to national regulations plasma from a female donor is forbidden, therefore we know that subjects were only adult men. Although specimens are anonymized, according to national regulations ("Spendezulassung B-CH SRK Nr. 07-21") donors must be asymptomatic for at least 28 days. Mean age and mean DPP/DPOS are given only for the 26/34 specimens for which patient data were available.

**Table 2 Characteristic of vaccinated individuals' samples.**

| Vaccine | Number of patient | Gender (M/F) | Age mean | WPV Mean weeks (range) | Sample type | Date of vaccination |
|---|---|---|---|---|---|---|
| mRNA vaccine | 16 | 4/12 | 52 | 8 | Serum, plasma | March–May 2021 |

*WPV weeks post 2nd dose vaccination.*

**Table 3 Characteristic of vaccinated and infected individuals' samples.**

| Vaccination/infection status | Number of patient | Gender (M/F) | Age mean | WPV Mean weeks (range) | Sample type | Interval vaccination –infection Mean weeks (range) |
|---|---|---|---|---|---|---|
| Prior infection + 2× vaccination | 6 | 3/3 | 46 | 4 (4–5) | Plasma | 33 (18–54)[a] |
| 2× vaccination + breakthrough (Delta) | 13 | 5/8 | 43 | 30 (9–43) | Serum | 22 (5–35) |
| 2× vaccination + breakthrough (Omicron-BA.1) | 8 | 4/4 | 38 | 28 (12–47) | Serum | 23 (8–41) |
| 1× vaccination + breakthrough (Omicron-BA.1) | 3 | 3/0 | 28 | 16 (5–37) | Serum | 12 (3–32) |

[a]3/6 individuals were infected in March, October, and December 2020.

change reduction of 2.2 for Alpha, 2.7 for Beta, 3.6 for Gamma, 3.1 for Delta, 25.9 for Zeta and 45.6 for Omicron-BA.1 (Fig. 1A). None of the samples failed to neutralize the homologous virus. Only 1/34 (3%), 2/34 (6%), 3/34 (9%) and 3/34 (9%) failed to neutralize Alpha, Beta, Gamma, and Delta, respectively, whereas 21/34 (62%) and 29/34 (85%) completely failed to neutralize the Zeta and Omicron-BA.1 variant.

For patients previously infected with the Alpha variant ($n = 12$), geometric mean $PRNT_{90}$ titers were 45.5 (95%CI: 34.3–60) for Alpha, 27.8 (95%CI: 19.8–39.0) for B.1, 7.4 (95%CI: 4.0–13.5) for Beta, 3.8 (95%CI: 1.6–8.7) for Gamma, 5.9 (95%CI: 2.8–12.2) for Delta, 1.6 (95%CI: 0.7–3.6) for Zeta and 0.8 (95%CI: 0.4–1.7) for Omicron-BA.1. Compared to the homologous Alpha variant, this results in a reduction of 1.6 (B.1), 6.2 (Beta), 12.0 (Gamma), 7.7 (Delta), 28.2 (Zeta) and 56.1 (Omicron-BA.1). Complete loss of neutralization was observed for 3/12 (25%) samples for Gamma, 1/12 (8%) for Delta, 5/12 (42%) for Zeta and 10/12 (83%) for Omicron-BA.1 (Fig. 1B).

For individuals previously infected with the Beta variant ($n = 8$), geometric mean $PRNT_{90}$ titers were 20.6 (95%CI: 6.8–62.6) for Beta, 6.0 (95%CI: 1.3–27.2) for B.1, 2.6 (95%CI: 0.4–14.9) for Alpha, 2.6 (95%CI: 0.4–15.8) for Gamma, 3.2 (95% CI: 0.7–15.3) for Delta, 1.7 (95%CI: 0.4–7.5) for Zeta and 0.9 (95%CI: 0.4–2.3) for Omicron-BA.1. Compared to the homologous virus (Beta), this results in a fold-reduction of 3.5 for B.1, 8.0 for Alpha, 7.8 for Gamma, 6.4 for Delta, 12.4 for Zeta and 23.0 for Omicron-BA.1. Complete loss of neutralization was observed for 1/8 (12.5) for B.1, 4/8 (50%) for Alpha, 4/8 (50%) for Gamma, 3/8 (37.5) for Delta, 4/8 (50%) for Zeta and 6/8 (75%) for Omicron-BA.1 (Fig. 1C).

For individuals previously infected with the Gamma variant ($n = 10$), geometric mean $PRNT_{90}$ titers were 55.6 (95%CI: 24.1–128) for Gamma, 20.5 (95%CI: 7.6–55.5) for B.1, 13.9 (95% CI: 5.9–32.9) for Alpha, 18.3 (95%CI: 8.9–37.4) for Beta, 3.2 (95% CI: 1.0–10.1) for Delta, 10.2 (95%CI: 5.2–20.2) for Zeta and 2.1 (95%CI: 0.7–6.4) for Omicron-BA.1. Compared to the homologous virus, fold reduction in neutralization was 2.7 for B.1, 4.0 for Alpha, 3.0 for Beta, 17.6 for Delta, 5.4 for Zeta and 26.9 for Omicron-BA.1. Complete loss of neutralization was observed for 4/10 (40%) for Delta and 5/10 (50%) for Omicron-BA.1 (Fig. 1D). Of note, a rather strong loss of neutralization in Gamma

convalescent samples was observed for Delta, while neutralization was less affected for the Zeta variant.

For individuals previously infected with Delta ($n = 10$), geometric mean $PRNT_{90}$ titers were 72.8 (95%CI: 33.9–156.2) for Delta, 25.1 (95%CI: 14.0–45.1) for B.1, 18.4 (95%CI: 9.5–35.4) for Alpha, 13.2 (95%CI: 7.4–23.5) for Beta, 15.0 (95%CI: 7.8–28.7) for Gamma, 10.4 (95%CI: 4.0–26.9) for Zeta and 3.1 (95%CI: 1.0–9.6) for Omicron-BA.1. Fold reduction compared to homologous virus (Delta) was 2.9 for B.1, 4.0 for Alpha, 5.5 for Beta, 4.9 for Gamma, 7.0 for Zeta and 23.8 for Omicron-BA.1. Complete loss of neutralization was 1/10 (10%) for Zeta and 4/10 (40%) for Omicron-BA.1 (Fig. 1E).

A heatmap for fold-change reduction of neutralization was generated to summarize the findings (Fig. 2). Here, rather robust neutralization of pre-VOC convalescent specimens against VOCs Alpha, Beta, Gamma, and Delta is visible while the other variants showed stronger escape from immunity against heterologous variants. Immune escape properties for Zeta and Omicron-BA.1 are visible. Zeta displayed stronger immune escape against the pre-VOC variant and Alpha than against Gamma and Delta. Immune escape of Omicron-BA.1 was pronounced throughout all specimens, although the fold-change reduction is in a comparable range to that of Zeta for some combinations (25.9- and 28.2-fold in pre-VOC and Alpha convalescent for Zeta, respectively, and 23.0- and 23.8-fold in Beta and Delta-convalescent for Omicron-BA.1, respectively).

**Neutralizing capacity from post-vaccine and combined post-vaccine/infection-derived samples.** We investigated a total of 46 patient specimens from either double-vaccinated individuals ($n = 16$) or with combined vaccination-infection-derived immunity, either through prior infection followed by vaccination, or vaccination followed by a vaccine-breakthrough infection with Delta or Omicron-BA.1 ($n = 30$).

In contrast to all infection-derived convalescent samples, geometric mean $PRNT_{90}$ titers were much higher for individuals double-vaccinated with either BNT162b2 or mRNA-1273 with titers of 338.0 (95%CI: 247.4–461.6) against B.1, 121.7 (95%CI: 86.0–172.3) against Alpha, 49.3 (95%CI: 28.1–86.8) for Beta, 62.8 (95%CI: 36.0–109.5) for Gamma, 95.6 (95%CI: 69.4–131.7) for Delta, 78.5 (95%CI: 50.4–122.5) for Zeta and 3.9 (95%CI: 1.8–8.7)

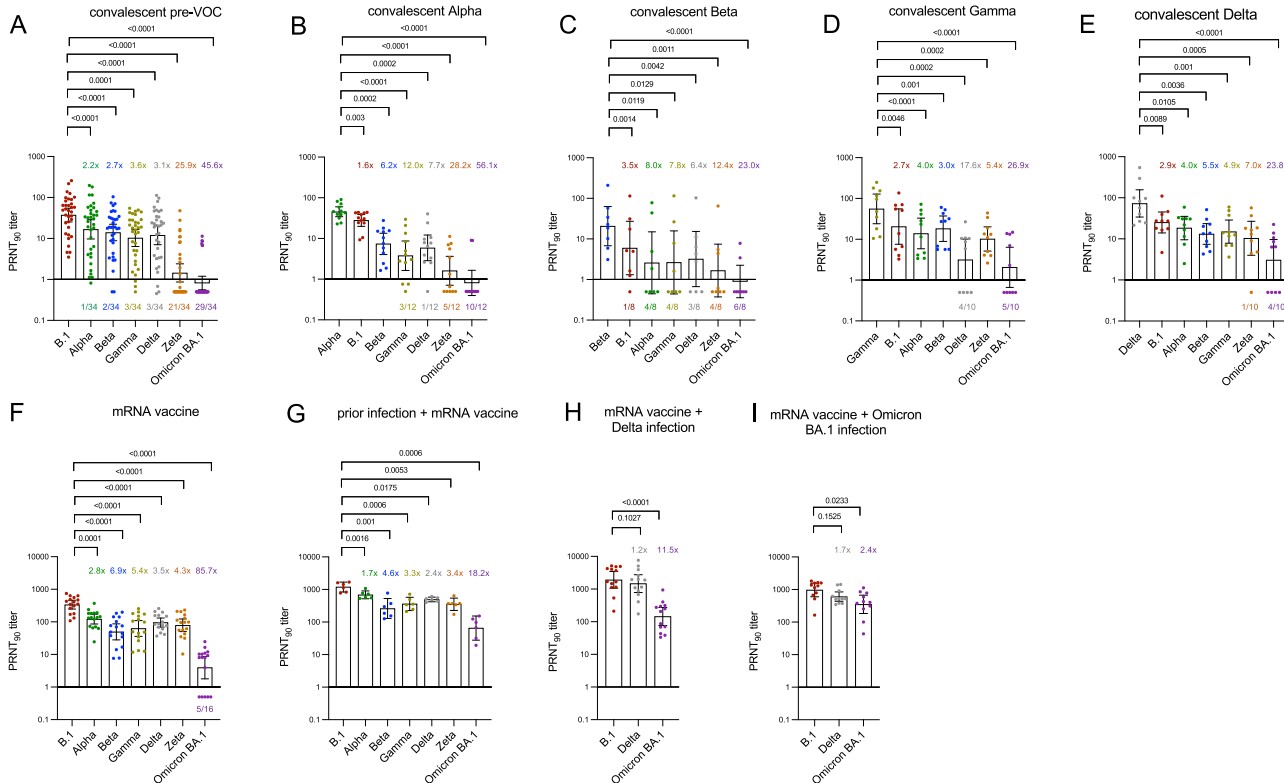

**Fig. 1 Neutralization in infection-derived blood specimens against seven authentic isolates of SARS-CoV-2 variants (B.1, Alpha, Beta Gamma, Delta, Zeta, Omicron-BA.1).** Bars represent geometric mean titers (GMT) of 90% reduction endpoint titers (PRNT$_{90}$) with 95% confidence interval. **A**–**E** Cohorts of convalescent specimens that are derived from individuals infected with **A** early-pandemic SARS-CoV-2 (pre-VOC), **B** Alpha, **C** Beta, **D** Gamma, and **E** Delta. **F**–**I** Cohorts consist of individuals with **F** double-dose mRNA vaccination, **G** prior SARS-CoV-2 infection followed by double-dose mRNA vaccination **H** Delta breakthrough infection of double-vaccinated individuals and **I** Omicron-BA.1 breakthrough infection following double (n = 8) and single (n = 3) mRNA vaccination. Colored numbers above bars refer to fold change reduction of GMT versus the homologous (infecting) variant, shown as first bar of each figure. Colored numbers below each bar represent the number of specimens with complete loss of neutralization (PRNT$_{90}$ titer < 1). Repeated measures one-way ANOVA with Dunnett's multiple comparisons test using log10 transformed PRNT$_{90}$ titers was performed to analyze the statistical significance. Source data are provided as a Source Data file.

for Omicron-BA.1. This translates into a fold reduction of neutralization of 2.8 for Alpha, 6.9 for Beta, 5.4 for Gamma, 3.5 for Delta, 4.3 for Zeta, and 85.7 for Omicron-BA.1. No complete loss of neutralization was seen for any VOC except Omicron-BA.1 in 5/16 (31%) specimens (Fig. 1F).

Individuals with prior SARS-CoV-2 infection before double vaccination (n = 6), as determined by anti-nucleocapsid antibody presence, showed geometric mean PRNT$_{90}$ titers of 1190.4 (95% CI: 837.8–1691) against B.1, followed by 683.2 (95%CI: 516.3–904.1) for Alpha, 260.4 (95%CI: 128.8–526.5) for Beta, 360.4 (95%CI: 224.5–578.5) for Gamma, 494.1 (95%CI: 419.4–582.1) for Delta, 351.8 (95%CI: 227–545.2) for Zeta and 65.2 (95%CI: 27.81–153.0) for Omicron-BA.1. Fold reduction in neutralization was 1.7 for Alpha, 4.6 for Beta, 3.3 for Gamma, 2.4 for Delta, 3.4 for Zeta, and 18.2 for Omicron-BA.1. Of note, none of the specimens showed complete loss of neutralization (Fig. 1G).

In addition, we have investigated vaccinated individuals with a breakthrough infection with Delta (n = 13) and Omicron-BA.1 (n = 11) for neutralization against both viruses which are currently the only VOCs co-circulating. For the first group, high geometric mean PRNT$_{90}$ titers of 1915 (95%CI: 1059–3463) were observed for B.1, 1472 (95%CI: 785–2760) were observed for Delta while 144.3 (95%CI: 76.49–272.1) was observed for Omicron-BA.1. This results in a 1.2-fold reduction for Delta and 11.5 for Omicron-BA.1 versus the homologous B.1, but no

complete loss of neutralization was observed. (Fig. 1H). For Omicron-BA.1 breakthrough infection following vaccination, geometric mean PRNT$_{90}$ titers of 966.9 (95%CI: 597.9–1564) were observed for B.1, 606.1 (95%CI: 434.3–845.7) were observed for Delta, which equals a 1.7-fold loss of neutralization to homologous B.1 and 351.3 (95%CI: 184.5–668.9) we observed against Omicron-BA.1 which equals a 2.4 fold reduction in neutralization to B.1 (Fig. 1I). Higher titers were seen against Delta compared to the infecting Omicron-BA.1.

A heatmap of fold-change reduction of neutralization summarizes the findings across all post-vaccine and combined infection/vaccine specimens (Fig. 2). The pronounced escape from vaccination specific for Omicron-BA.1 is visible, while much less immune escape was observed for the other VOCs as well as for Zeta. Neutralization of Omicron-BA.1 is improved in all specimens with combined infection/vaccination immunity.

We also performed a mapping of our titration results using antigenic cartography (Fig. 3). Here we could show that homologous sera cluster around the respective infecting virus, with Alpha and pre-VOC specimens clustering together most closely. Earlier variants of concern before Omicron-BA.1 (Alpha, Beta, Gamma, Delta) belong to one antigenic cluster. Zeta and Omicron-BA.1 are more distantly represented in the map with more than 3 units distance to all other viruses, thus presenting two separate antigenic clusters (Fig. 3A). Post vaccine sera cluster around B.1 and Alpha strains, and a larger but equal distance to

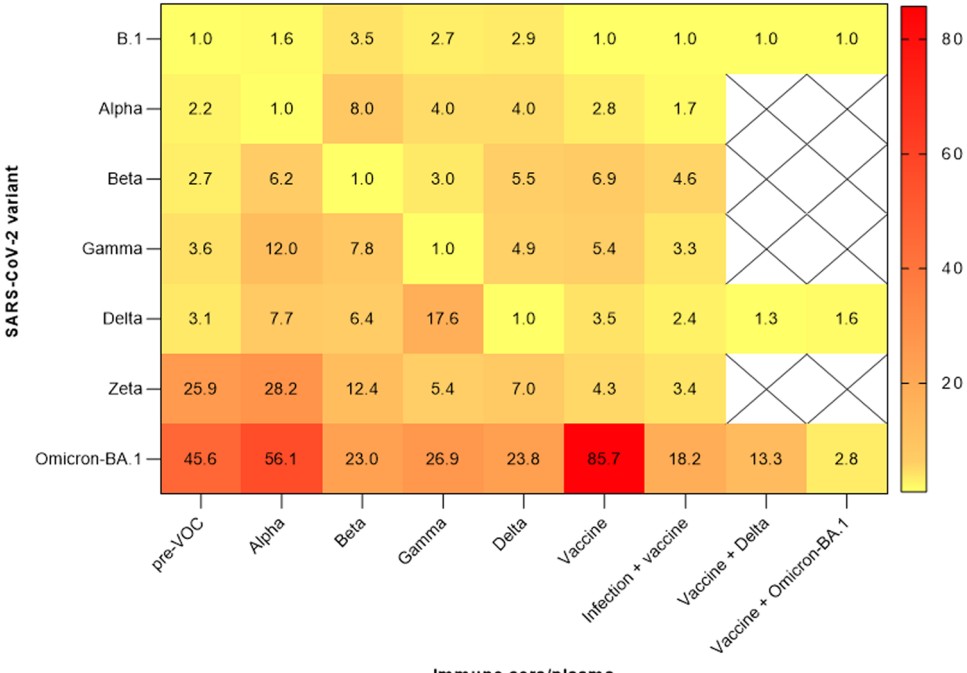

**Fig. 2 Heatmap of fold-reduction in neutralization based on PRNT$_{90}$ data. Values of fold-reduction in neutralization (PRNT$_{90}$) of B.1, Alpha, Beta, Gamma, Delta, Zeta, and Omicron-BA.1 are presented as heat maps with darker colors implying greater changes.** The immune sera/plasma were organized into cohorts of convalescent specimens that are derived from individuals infected with early-pandemic SARS-CoV-2 (pre-VOC) ($n = 34$), Alpha ($n = 12$), Beta ($n = 8$), Gamma ($n = 10$), Delta ($n = 10$) and cohorts consist of individuals with double-dose mRNA vaccination ($n = 16$), prior SARS-CoV-2 infection followed by double-dose mRNA vaccination ($n = 6$), Delta breakthrough infection of double-vaccinated individuals ($n = 13$) and, Omicron-BA.1 breakthrough infection following double ($n = 8$) and single ($n = 3$) mRNA vaccination. Source data are provided as a Source Data file.

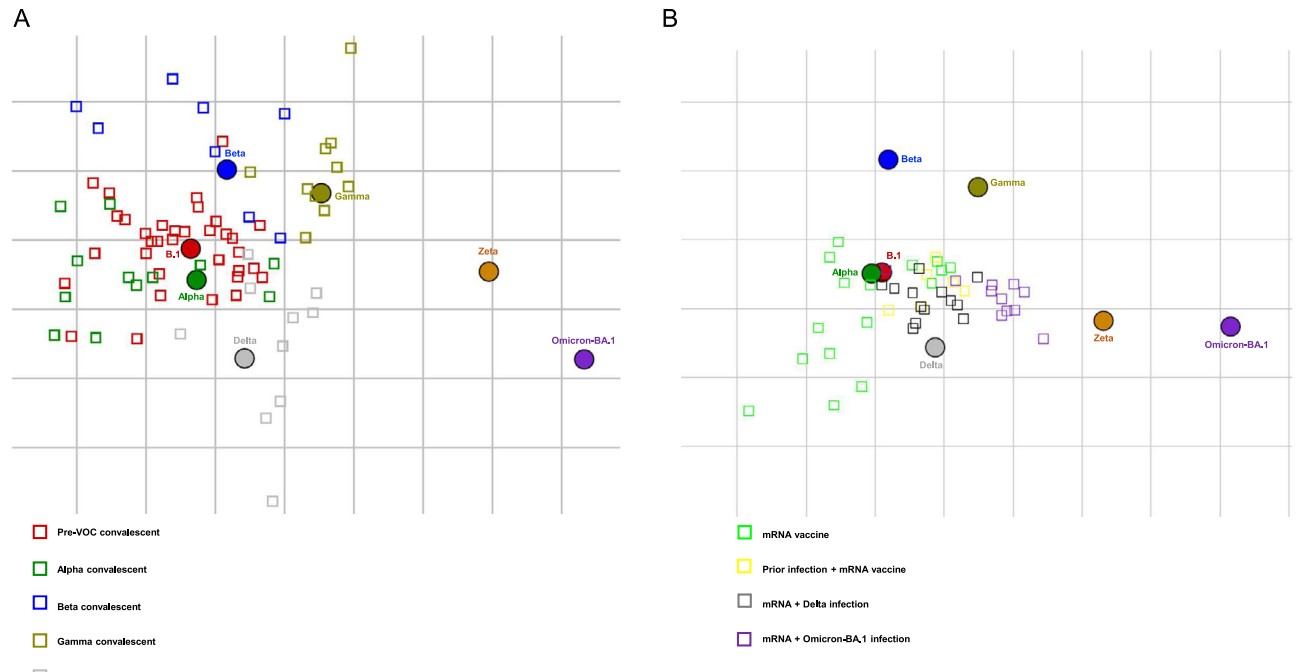

**Fig. 3 SARS-CoV-2 antigenic cartography.** Mapping of antigenic relationship using antigenic cartography for **A** convalescent specimens from pre-VOC SARS-CoV-2, Alpha, Beta, Gamma, and Delta infections and **B** post-vaccination specimens with and without prior infection and breakthrough infections with Delta or Omicron-BA.1. For graphical reasons only post vaccination specimens are shown in (**B**). Source data are provided as a Source Data file.

Beta and Gamma is observed (Fig. 3B). Again, Zeta shows a larger antigenic distance, but Omicron-BA.1 is the most distant. Of note, specimens with combined immunity from infection followed by vaccination are located closer to the other non-

alpha VOCs compared to post-vaccination samples without prior infection. Also, sera from individuals with breakthrough infection with Delta or Omicron-BA.1 are located either between B.1/Alpha and Delta VOCs or more centrally between B.1/Alpha,

Delta, and Omicron-BA.1, indicating a broader neutralizing response.

**Individual profiles of SARS-CoV-2 homologous and heterologous neutralization**. In addition to accumulated data (Figs. 1 and 2), neutralization profiles were also displayed on an individual basis (Figs. S2–S10). Here, inter-individual differences both in the quantity of antibody response as well as patterns of neutralization loss towards heterologous variants were observed, which is not necessarily reflected in the pooled results of all individuals. For example, while neutralization against Omicron-BA.1 was the lowest throughout convalescent samples, the differences between Alpha, Beta and Gamma were less pronounced in some individuals. Of note, in all individual comparisons, the most efficient neutralization was always observed against the homologous (infecting) strain through all convalescent sera as well as vaccine sera.

**Correlation of receptor-binding-domain (RBD) binding antibodies with neutralizing titers**. In order to understand whether RBD-binding IgG antibodies, as often measured in clinical laboratories, are a good predictor of neutralizing responses against VOCs in the context of different immunity backgrounds, we correlated PRNT90 titers with RBD-binding titers using a commercially available multiplex assay that determines RBD-binding IgG antibody titers against ancestral SARS-CoV-2 and all VOCs (Alpha, Beta, Gamma, Delta, Omicron-BA.1). Due to the low number of sera that could be analyzed for the groups of Alpha convalescent and prior-infection + vaccination individuals, we did not perform correlation analysis for these groups.

For convalescent pre-VOC sera/plasma ($n = 34$), pre-VOC RBD IgG antibodies correlated well with neutralization against B.1, Alpha, Beta, Gamma Delta, and Zeta (Spearman's $\rho = 0.7$–$0.88$) but only moderately with Omicron-BA.1 (Spearman's $\rho = 0.51$). Even Omicron RBD-binding IgG antibodies did not show better correlation with Omicron-BA.1 PRNT90 titers (Spearman $\rho = 0.59$) (Fig. S11A).

For convalescent Beta sera ($n = 8$), pre-VOC RBD showed a moderate to good correlation with neutralizing titres against all VOCs, but the correlation was only significant for Alpha, Gamma, and Omicron PRNT90 titres. Using Beta RBD as antigen let to slightly higher correlation coefficients and correlation with B.1 and Beta PRNT90 titres became significant as well (Fig. S11B).

For convalescent Gamma sera ($n = 8$), preVOC RBD IgG antibodies correlated only moderately to low with neutralization against VOCs (Spearman's $\rho = 0.20$–$0.67$) but correlations were not significant. Using Gamma RBD as an antigen did not improve correlation substantially (Fig. S11C).

For convalescent Delta ($n = 10$), pre-VOC RBD correlated significantly with neutralization against Delta (Spearman's $\rho = 0.76$) but not against other variants. Using Delta RBD as antigen let to slightly higher correlation coefficients against neutralization of all VOCs and correlation with B.1 and Omicron PRNT90 titres became significant as well (Fig. S11D).

In post-vaccine sera/plasma ($n = 16$), pre-VOC RBD IgG antibodies correlated moderately with neutralization against B.1, Alpha, Delta and Omicron-BA.1 (Spearman's $\rho = 0.53$–$0.62$) but correlated better with neutralization against Beta, Gamma, and Zeta (Spearman's $\rho = 0.82$–$0.87$). Nevertheless, all correlations were significant. Using RBD of other variants did not improve correlation against neutralization titres (Fig. S12A).

In vaccinated individuals with a breakthrough infection with Delta ($n = 13$), pre-VOCs and Delta RBD IgG antibodies correlated moderately but significantly with neutralization against

B.1 and Delta (Spearman's $\rho = 0.57$–$0.58$). Correlation with Omicron-BA.1 was lower and not significant (Spearman's $\rho = (0.34$–$0.35)$ (Fig. S12B). Using Omicron RBD-binding IgG antibodies did not improve correlation with Omicron-BA.1 PRNT90 titers (Spearman $\rho = 0.40$) (Fig. S12B).

For Omicron-BA.1 breakthrough infection following vaccination ($n = 11$), there is only low and non-significant correlation between pre-VOC RBD IgG antibodies and neutralization against B.1, Delta, and Omicron-BA.1 (Spearman's $\rho = 0.2$–$0.25$). Using Omicron RBD as antigen improved correlation against Omicron neutralization titres (Spearman's $\rho = 0.44$), but was not significant as well (Fig. S12C).

## Discussion

Here we assess neutralizing capacity towards seven SARS-CoV-2 variants by convalescent specimens from individuals recovered from: infection with the early-pandemic strain (pre-VOC); the VOCs Alpha, Beta, Gamma, or Delta; from double-vaccinated individuals, either with or without prior infection; and from double-vaccinated individuals infected with the Delta or Omicron-BA.1 variant.

We showed that the highest SARS-CoV-2 neutralizing titers, either elicited through infection or vaccination, were always observed against the homologous strain (infecting strain or antigen used in the vaccine formulation), whereas a reduced neutralizing capacity was found for heterologous strains.

In agreement with our results, one study found lower neutralization of the Alpha and Gamma variant by pre-VOC convalescent sera and lower neutralization of pre-VOC and Alpha in Gamma infected patients, but reduced neutralization of only Gamma in vaccine sera[21]. In line with these observations, Alpha-infected individuals showed reduced neutralization of the Beta variant[22]. The reduction in the neutralization of the Delta variant was estimated to be between four- and eightfold in vaccinated sera, and sixfold in convalescent sera[23]. In addition, one pre-print investigated the neutralization of various VOCs by convalescent samples from individuals infected with pre-VOC, Alpha, Beta or Delta samples, and found the neutralization was strongest in the homologous convalescent-variant pairs for Alpha, Beta and Delta[24].

We observed reduced neutralization capacity of pre-VOC convalescent and post-vaccine samples against Alpha, Beta, Gamma, and Delta similar to what was described by others[23,25–28]. In comparison with neutralizing activity of other convalescent samples, pre-VOC sera showed rather robust neutralization against Alpha, Beta, Gamma, and Delta, while VOC-convalescent sera showed lower potential to neutralize heterologous viruses. While for the Alpha variant, only slightly reduced neutralization was described for both convalescent and vaccinee sera, a more pronounced reduction of neutralization was observed for the Beta and Gamma variants[26,27,29–32]. Thus, Beta and Gamma comprised the two VOCs with the most pronounced immune escape and ability to cause large outbreaks in defined geographic regions (without reaching global dominance) before the emergence of Omicron. A recent study with pre-VOC SARS-CoV-2, Alpha, and Beta in multiple animals and cell culture models showed enhanced fitness of Alpha but not Beta, which could be a reason why Alpha, and not Beta, reached wide dominance in early 2021[33].

Across our panel of convalescent and vaccine sera, the strongest decline of neutralization capacity was observed for Zeta and Omicron-BA.1. While convalescent specimens showed strong reductions of neutralization of Zeta of up to 25.9-fold, neutralization capacity was restored in vaccinated individuals that showed only 4.3-fold decline compared to pre-VOC SARS-CoV-2. In contrast, for

Omicron-BA.1, a stark loss of neutralizing activity of up to 46-fold or 86-fold was observed for convalescent and vaccine sera, respectively. Thus, the fold-change reduction was even higher in vaccine sera than in convalescent sera, although titers in vaccine sera were higher and therefore the percentage of specimens with complete loss was lower (31% of specimens vs. 83%). A similar loss of neutralization capacity against Omicron-BA.1 was observed in other studies[24,34–36], although no validation across the full range of VOC convalescent sera was available before our data set. Interestingly, the strongest loss of neutralization for Zeta was in a similar range to the weakest loss of neutralization for Omicron-BA.1, e.g. a 25.9-fold reduction of Zeta in pre-VOC convalescent samples, and a 23.0-fold reduction of Omicron-BA.1 in Beta convalescent samples. Such observed differences in fold-change reduction between Zeta and Omicron-BA.1 were less than twofold in pre-VOC, Alpha, and Beta convalescent sera, showing that even before the emergence of Omicron-BA.1, variants with strong immune escape properties almost reaching that of Omicron-BA.1 were already circulating but did not become dominant. For post-Gamma and post-Delta specimens, the differences between fold-change reduction of Zeta and Omicron-BA.1 were much bigger, with Omicron-BA.1 showing a 3-5-fold higher escape of neutralization compared to Zeta. At least for Gamma-convalescent sera, robust neutralization of Zeta could be explained by a common origin as both are descendants from B.1.1.28 (Pango lineage)[10]. To the best of our knowledge, no data on neutralization of the Zeta variant by convalescent or vaccinee samples have been described to date, and few other data were obtained on this variant. Loss of neutralization of Delta was much more pronounced in Gamma-convalescent sera than in any other convalescent or vaccine sera with a 17.6-fold reduction. Such a strong immune escape of Delta in a variant-specific infection background has not been reported before.

A recent preprint mapped antigenic diversity of SARS-CoV-2, a method that was originally developed to map antigenic relationship for influenza viruses by hemagglutinin inhibition assay[37,38]. When used to assess influenza viruses, one unit represents a twofold change in neutralization titer, and a distance of more than 3 units is required for a separate antigenic cluster, while below 3, it is considered antigenically similar. They show that ancestral SARS-CoV-2, Alpha, Beta, Gamma, and Delta form one antigenic cluster, while Omicron-BA.1 forms a separate antigenic cluster. In contrast to our study, neutralization was only done with a pseudovirus assay, and with a lower number of convalescent samples from patients infected with VOCs. Upon analysis for antigenic cartography, our data confirm the findings of Omicron-BA.1 as another antigenic cluster by authentic virus PRNT, and additionally, we are the first to show Zeta forms a separate antigenic cluster[38]. Furthermore, we show that combined immunity after infection/vaccination shows a reduced distance to heterologous variants such as Beta, Gamma, and Delta in the antigenic map when compared to vaccine samples without additional infection, indicative of broader neutralization.

Of note, we have observed large differences in neutralization capacity across specimens with different infection backgrounds, which has not been described in this detail before. While convalescent specimens of individuals infected with a pre-VOC SARS-CoV-2 and Alpha showed 45.6- and 56.1-fold reduction of neutralization capacity for Omicron-BA.1, patients previously infected with the Beta, Gamma or Delta variant showed a lower reduction in neutralization capacity of 23.0, 26.9 and 23.8-fold, and a lower percentage of specimens with complete neutralization failure, respectively. These differences could indicate that regional heterogeneity in background immunity could potentially influence emergence and spread of Omicron-BA.1 or other future variants with immune escape properties.

A gradient of immune escape can be described from the variants investigated here, from pre-VOC to Alpha (only slight immune escape, but successfully outcompeted earlier strains), Beta and Gamma (more pronounced immune escape with regionally pronounced circulation but no global dominance) to Zeta (variant with one of the strongest escapes of immunity prior to Omicron-BA.1, but limited transmission). Our data on Zeta, along with data two other VOIs Mu and Lambda, have shown the strongest escape from neutralization in the pandemic period before the emergence of Omicron-BA.1[39–42]. Although differences between escape from neutralization between Zeta and Omicron-BA.1 are only two-fold in some subgroups, neutralization for Zeta was restored in vaccinee specimens, while this was not the case for Omicron-BA.1. This could hint at different mechanisms of immune escape between SARS-CoV-2 variants, and other fitness advantages for Omicron-BA.1 beyond immune escape.

Of note, a mutation of position 484 in the receptor-binding domain is found in Beta, Gamma, Zeta (E484K), and Omicron-BA.1 (E484A). The same mutation arose independently also in a lineage of the Alpha variant where it was also associated with escape from neutralizing antibodies[43]. It has been shown that a mutation at position 484 of the spike tends to have the strongest effect on receptor binding and neutralization[44–47]. While the mutation can explain, at least partly, the strong escape from neutralization in Omicron-BA.1, it has no or only very low influence on ACE2-binding and is, therefore, most likely not associated with higher transmissibility, although the mechanism of transmissibility in Omicron-BA.1 is not well understood[48].

We have shown that Omicron-BA.1 exhibits a strong escape from neutralization in both convalescent and vaccine sera, although differences in fold change reduction exist depending on prior infection background. However, it has been shown that a third dose of a mRNA vaccine is able to restore neutralizing capacity[49–52]. Similarly, individuals with mixed immunity, i.e., infection prior to double vaccination, or infection after double vaccination, also leads to higher neutralizing capacity towards Omicron-BA.1[24,36]. With an increasing number of vaccine breakthrough infections observed during the Delta wave and reports of a large number of vaccine breakthrough infections with Omicron-BA.1, an assessment of mixed immunity is of huge interest, especially for protection against new variants[53]. In addition, it has been speculated that the immune response towards an antigenically divergent SARS-CoV-2 variant will be influenced by pre-existing immunity[54].

Here we investigated 13 individuals with a Delta breakthrough infection after double vaccination and have found high neutralizing titers for Delta. Neutralization capacity for Omicron-BA.1 was markedly reduced, but titers were still higher than titers against other VOCs in double-vaccinated individuals indicating that even a boost with a mismatched strain (Delta) can lead to a considerable increase in immunity against Omicron-BA.1. Similarly, vaccine breakthrough infections with Omicron-BA.1 (n = 11) resulted in very high neutralization titers against the Omicron-BA.1 variant, indicating that infection with an antigenically dissimilar variant led to a robust Omicron-BA.1-specific immune response despite the presence of pre-existing immunity against the original pandemic strain. Interestingly, neutralizing titers against the Delta variant were roughly comparable to Omicron-BA.1 indicating that infection with antigenically different variants can boost immunity against variants that are antigenically similar to the vaccine strain.

Currently, one of the most widely used serological assays uses the RBD of the original Wuhan spike from the pre-VOC period of the pandemic as antigen, in some instances also with the aim to determine protection. Since nowadays the ancestral SARS-CoV-2 is not circulating anymore but was replaced by antigenically different VOCs leading to a large variety of mixed immunity

backgrounds, correlation between RBD binding titers of such assays and neutralizing titers, as determined here, are of interest. A recent publication has shown that assessment of RBD binding antibodies using the Wuhan S1 sequence may only be a poor predictor for neutralizing immunity against VOCs[55]. Similarly, our results show that in individuals with Omicron breakthrough infections pre-VOC RBD titers were only a poor predictor of neutralization activity. In contrast, in Delta breakthrough infections pre-VOC RBD titers showed moderate to good correlation with neutralization against B.1 and Delta but not Omicron. For individuals that have been infected by only one SARS-CoV-2 variant or have only been vaccinated, pre-VOC RBD was generally a good predictor of neutralizing activity against the infecting and non-infecting variants, with some exceptions such as Beta and Gamma neutralization in Delta infected individuals. However, care should be taken when interpreting these results as for some combinations the number of sera analyzed was low. In summary, pre-VOC RBD is a moderate to good predictor of the neutralizing response as long as the infecting variant or the antigen used in the vaccine belongs to the same serotype. Of note, while the correlation between pre-VOC RBD and neutralization of variants is still good, the predicted neutralization potency of against heterologous variants is generally lower compared to the homologous variant.

Population immunity and evolutionary pressure might differ on a regional scale depending on the immunity induced by earlier circulating variants[56]. Although the origin of Omicron-BA.1 is not well understood, it was speculated that a high background of Beta-variant immunity has favored the development of Omicron's immune-escape properties[57]. In contrast to earlier variants, Omicron-BA.1 is showing a rapid increase in cases with a short doubling time, in addition to its immune-escape properties[58]. This, together with reduced neutralization, is also suggestive of decreased vaccine effectiveness, and will further complicate the management of the pandemic.

Limitations of our study are the relatively low number of specimens that were available for convalescent specimens of patients previously infected with variants. Furthermore, we cannot exclude that individuals infected with a VOC were not previously infected with a first-wave virus, and thus their antibodies are derived from multiple infections with more than one variant. In addition, the Beta variant, which could not be readily isolated on Vero-E6 cell lines had to be adapted to Vero-E6 in order to be usable for our PRNT, thus the isolate accumulated mutations, could affect the neutralization results. No convalescent samples were available from individuals infected with Zeta, and no convalescent samples were yet available from individuals infected with Omicron-BA.1 without prior vaccination. For the correlation analysis between RBD-binding antibodies and neutralizing titers, samples number were low, and therefore the reliability of the findings is low. Furthermore, testing of single time points after infection/vaccination can only provide a snapshot and not inform on the duration of antibody responses over time. All our blood specimens were collected at rather early time points after infection or vaccination, thus with time, a broader neutralizing capacity towards the heterologous variant might be observed due to somatic hypermutation and affinity maturation, although in the case of Omicron-BA.1 the most likely will not restore neutralization loss[59].

Overall, we could show that responses to variants before Omicron-BA.1 were associated with reduced, but not complete loss in neutralization both by infection-derived as well as vaccine-derived immunity. Variants such as Zeta have already shown immune escape properties, but did not become dominant at the time of their emergence. In contrast, Omicron-BA.1 displays strong immune escape with a surge in case numbers in many areas of the world. Whether immunity of a population was obtained by a vaccine or by exposure to different variants could alter the selection pressure and lead to the emergence of new

variants according to the source of the immunity. Reassuringly, a combination of vaccine/infection-derived immunity leads to broader antibody responses, which could enable the transition to an endemic situation.

## Methods

**Setting**. The University Hospitals of Geneva (HUG) are a tertiary center that runs outpatient testing for SARS-CoV-2 and hosts the national reference center for emerging viruses at the laboratory of virology. The laboratory is participating in and coordinating the SARS-CoV-2 variant and genomic surveillance funded by the Swiss Federal Office of Public Health with ongoing full genome sequencing of SARS-CoV-2 positive patient samples obtained through the diagnostic unit of our Centre[60]. Up to 400 positive specimens with a cycle threshold (Ct) < 32 have been sequenced each week since March 2021. From each variant that falls into any of the categories of VOC/VOI, at least one virus isolate is generated. Furthermore, access to convalescent specimens and post-vaccination specimens from individuals vaccinated and/or diagnosed at the Hospital are available through ongoing studies.

**Patient samples**. Convalescent sera during the early pandemic period (pre-VOC) were collected in the context of an ongoing prospective observational study at the Geneva University Hospitals (HUG) (Ethics approval number: CCER 2020-00516). In the main study, consecutive sampling of multiple sample types was performed, however, for the purpose of our study, only one convalescent serum per patient of a single collection time point was used. In addition, anonymized left-over plasma from apheresis collection of plasma (all collected in 2020) were available under the general informed consent of the University Hospitals of Geneva.

Convalescent sera obtained from patients infected with a SARS-CoV-2 VOCs (Alpha, Beta, Gamma, Delta, Omicron-BA.1) were collected in 2021 by contacting patients with confirmed SARS-CoV-2 infection for a blood collection in the convalescent period for the purpose of this study, and the same was done for vaccine breakthrough infections (Ethics approval number: CCER 2020-02323). For each patient infected with a variant, information on the infecting virus was available by full-genome sequence. No sequence information on the infecting strain was available from patients infected in 2020, however, the first SARS-CoV-2 VOC in Switzerland (Alpha) was only observed on Dec 24, 2020, and all pre-VOC samples were collected before that date. Plasma samples from vaccinated healthy individuals, vaccinated with two doses of BNT162b2 (Pfizer/BioNTech) or mRNA-1273 (Moderna) vaccine at 28-day intervals were available from a prospective observational study, collected 30 days after the 2nd dose (Ethics approval number: CCER ICOVax 2021-00430). Written informed consent was obtained from all adult participants, and from the legally appointed representatives (parents) of all minor participants except for anonymized left-over materials from blood donors. All approvals were obtained from the Cantonal Ethical board of the Canton of Geneva, Switzerland (Commission Cantonale d'Ethique de la Recherche).

Convalescent samples were only collected from individuals with an RT-PCR-confirmed diagnosis of SARS-CoV-2 in our diagnostic unit, and sera/plasma were collected 3–137 days after diagnosis or symptom onset (days post diagnosis, DPP). All vaccinated individuals were additionally tested for the presence of anti-nucleocapsid antibodies (Elecsys® Anti-SARS-CoV-2 anti-N) to screen for unrecognized infection prior to vaccination. There are no differences to be expected in the PRNT regarding the use of plasma or serum, therefore both sample types were used in parallel and are termed "convalescent samples".

**Viruses and cells**. Vero-E6 (ATCC CRL-1586) (provided by V. Thiel lab) and Vero E6-TMPRSS (provided by National Institute for Biological Standards and Controls) cells were cultured in complete DMEM GlutaMax I medium supplemented with 10% fetal bovine serum, 1x non-essential amino acids, and 1% antibiotics (Penicillin/Streptomycin) (all reagents from Gibco, USA). Vero-TMPRSS were received from the National Institute for Biological Standards and Controls (NIBSC, Cat. Nr. 100978).

All SARS-CoV-2 viruses used in this study were isolated from residual nasopharyngeal swabs collected from patients presenting at the HUG under general informed consent of the hospital that allows the usage of anonymized left-over materials. All patient specimens from which isolates were obtained were fully sequenced (Table S1). The following viruses were isolated as follows: B.1, Alpha, Gamma and Zeta were isolated and propagated in Vero-E6. The Beta variant isolate was initially isolated on A549 cells expressing human ACE2 (provided by Prof. M. Schmolke) before passaging on Vero-E6 cells[33]. Briefly, no primary Beta isolate could be obtained on Vero-E6 but only on A549 cells overexpressing human ACE2[61]. Therefore, after primary isolation, A549-hACE2 cells were mixed with Vero-E6 in a 1:1 ratio and inoculated with the passage 1 isolate. The next passage was done on Vero-E6 to generate the virus stock. The Omicron-BA.1 variant was primarily isolated on Vero-TMPRSS cells, then transferred to Vero-E6 for generation of virus stock. All virus stocks were titrated on Vero-E6 cells and fully sequenced. Sequences of initial patient specimens and obtained virus isolates were compared for acquired mutations (Fig. S1). Experiments with live infectious SARS-CoV-2 followed the approved standard operating procedures of our BSL-3 facility.

**Plaque reduction neutralization test (PRNT)**. Following the PRNT procedure, Vero-E6 cells were seeded at a density of $4 \times 10^5$ cells/mL in 24-well cell culture plates. A total of 34, 12, 8, 10, 10, and 46 sera/plasma samples from unvaccinated patients infected with pre-VOC SARS-CoV-2, Alpha, Beta Gamma, Delta or sera from individuals vaccinated with BNT162b2/mRNA-1273 with or without prior infections, respectively, were used for determining the neutralizing titers against B.1 (first pandemic wave strain containing only the D614G substitution in Spike), Alpha, Beta, Gamma, Zeta, Delta and Omicron-BA.1 variants as described earlier[62]. Briefly, all sera/plasma were heat-inactivated at 56 °C for 30 min and serially diluted in Opti-Pro serum-free medium starting from 1:10 until up to 1:5120 if necessary. Sera/plasma were mixed with 50PFU of variant isolates (B.1, Alpha, Beta, Gamma, Delta, Zeta, and Omicron-BA.1) and incubated at 37 °C for 1 h. All samples were run in duplicate, and for each neutralization experiment, an infection control (no serum/plasma) and a reference serum were used to ensure reproducibility between different experiments. Vero-E6 cells were washed 1× with PBS and inoculated with the virus serum/plasma mixture for 1 h. Afterward, the inoculum was removed and 500 μL of the overlay medium also used for the plaque assays was added. After incubation for 3 days at 37 °C, 5% $CO_2$, the overlay medium was removed, cells were fixed in 6% formaldehyde solution for at least 1 h, plates were washed 1× with PBS and stained with crystal violet. Plaques were counted in wells inoculated with virus-serum/plasma mixtures and compared to plaque counts in infection control wells. The 90% reduction endpoint titers ($PRNT_{90}$) were calculated by fitting a 4-parameter logistics curve with variable slope to the plaque counts of each serum/plasma using GraphPad Prism version 9.1.0. For samples that did not reach 90% reduction at a 1:10 dilution, we extrapolated the titer until a dilution of 0.5. If the extrapolation reached a titer below 0.5, the sample was given a value of 0.5. All samples with a titer below 1, i.e. undiluted sample are considered negative.

**Antigenic cartography**. Antigenic maps were constructed with the online tool at https://acmacs-web.antigenic-cartography.org. The maps were generated with $PRNT_{90}$ titers obtained for convalescence specimens from pre-VOC SARS-CoV-2, Alpha, Beta, Gamma and Delta and for post-vaccine specimens with and without prior infection and breakthrough infections with Delta and Omicron-BA.1. First, all titers were rounded to the next whole number, and titers below 1 were designated as <1. Maps were constructed in two dimensions with 500 optimizations and the minimum column basis was set to "auto"[38].

**Multiplex antibody assay**. All sera/plasma samples used in this work, with the exception of 8 sera out of a total of 120 for which no sufficient volume was left, were analyzed by V-PLEX SARS-CoV-2 assay (Meso Scale Diagnostics, Gaithersburg, USA) panel 22 that detects and quantifies anti-SARS-CoV-2 IgG specific for Spike-RBD protein against 6 SARS-CoV-2 VOC (pre-VOC, Alpha, Beta, Gamma, Delta, and Omicron-BA.1) according to the manufacturer's instructions[63,64].

**Statistical analysis**. Data collection was done using Excel 2019. Geometric means with 95% CI were used for the comparison of $PRNT_{90}$ titers. Statistical analyses for $PRNT_{90}$ were conducted using GraphPad Prism version 9.1.0 software and performed using repeated measures one-way ANOVA with Dunnett's multiple comparisons test with log10 transformed $PRNT_{90}$ titers.

Correlation statistical significance was assessed by two-sided Spearman correlation using R statistical software version 4.1.1; *p values < 0.05, **p values < 0.01, ***p values < 0.001, ****p values < 0.0001.

**Reporting summary**. Further information on research design is available in the Nature Research Reporting Summary linked to this article.

## Data availability

The minimum datasets presented in this article are included as a Source Data file in this manuscript. The personal information related to the participants presented in this article are available under restricted access because no consent has been obtained from participants for public sharing of anonymized datasets. Requests to access the personal information of participants should be directed to corresponding authors. Source data are provided with this paper.

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

## Acknowledgements

We thank Pascale Sattonnet-Roche and Barbara Lemaitre for their excellent technical help. We thank the staff of the laboratory of virology at the HUG for support. We thank all clinicians and technical staff responsible of the different clinical cohorts for their help. We are grateful for the patients who were willing to donate their samples and agree to participate in our research. We thank Samuel Cordey and Florian Laubscher for help with sequence analysis. We thank Mirco Schmolke and Beryl Mazel-Sanchez for A549-hACE2 cells. We thank Volker Thiel, Jenna Kelly and Silvio Steiner, Vetsuisse Bern, for help with Omicron-BA.1 sequencing. This work was supported by the Swiss National Science Foundation 196644 (IE), 196383 (IE), NRP (National Research Program) 78 Covid-19 Grant 198412 (IE, BM), the Fondation Ancrage Bienfaisance du Groupe Pictet (IE) and the Fondation Privée des Hôpitaux Universitaires de Genève (IE).

## Author contributions

M.B. and K.A. performed the experiments and conducted the analysis, P.V., C.S.E., K.H.F., D.V., O.P., S.W.A., C.S., A.G.L., C.A.S., A.M.D., and L.K. conducted the clinical studies and/or helped with clinical sample collection, M.E.L. contributed to virus isolation of variants and isolate characterization. B.M. and I.E. designed and supervised the study. M.B., I.E., and B.M. wrote the main manuscript. All authors have contributed to the final version of the manuscript.

## Competing interests

Financial: A.D. is a consultant for Speranza, Non-financial: Chairman for WHO Technical Advisory Group (TAG) on Emergency Use Listing of COVID-19 vaccines. All other authors declare no competing interests.
