## [Peer Review File · Nature Communications]

Neutralization capacity of antibodies elicited through homologous or heterologous infection or vaccination against SARS-CoV-2 VOCsReviewers' Comments:

Reviewer #1:

Remarks to the Author:

Bekliz et al. determined the ability of sera/plasma from convalescents infected with ancestral SARS-CoV-2, Alpha, Beta, Gamma or Delta to neutralise a panel of variants of concerns using plaque reduction neutralisation assay. Similar analysis was performed with sera/plasma from individuals vaccinated with 2 doses of mRNA vaccine with or without pre-infection and from vaccinees after a breakthrough infection with Delta or Omicron. Highest neutralization titers were observed against the homologous variant. The greatest immune escape was detected with Omicron. This immune escape was reduced in individuals who experienced a pre-infection before a double vaccination. Breakthrough infection with Omicron induced antibodies also able to neutralise Delta. The authors also performed an antigenic cartography of variants which identified Zeta and Omicron as separate antigenic clusters.

This paper is very well written, clear and transparent. This paper is not the first one which compares heterologous humoral immunity but to my knowledge the other ones either compared less combinations or they used pseudoviruses (e.g. 10.1101/2022.01.28.477987, 10.1101/2022.01.03.21268582, 10.1126/science.abm0811). The plaque reduction neutralization assay is considered as the gold standard assay to determine neutralising responses. Data look accurate to me and robust. In addition, I had not seen any neutralisation data using Zeta isolate before this paper. The limitations are also well mentioned including the fact it is not a longitudinal study.

Some preprints had shown that antibodies induced by an infection with Beta could neutralise better Gamma than Beta but they used pseudovirus neutralization assays (10.1101/2022.01.28.477987, 10.1101/2022.01.03.21268582). The results observed in this paper make more sense to me.

Please find below some comments to potentially strengthen the paper.

Major comments:

1) Do the authors have any RBD-specific IgG binding data ? Did the authors analyse the relationship between VOC RBD binding IgG and neutralization responses ? A paper which analysed vaccine-induced neutralising responses against VOCs in individuals with or without pre-infection with Alpha showed that S1 serology data using the Wuhan Hu-1 S1 RBD and VOC sequence was an unreliable marker for neutralization potency against variants of concerns (10.1126/science.abm0811). I think it would be valuable to compare the correlations between IgG binding data and neutralising responses in the context of the different combinations analysed in this paper.

2) Do the authors have similar data with mucosal samples (saliva, nasal samples) ? The comparison of systemic and mucosal neutralising responses would be very informative.

3) The authors analysed the neutralising capacity of sera from vaccinees who were pre-infected before vaccination. The type of variant which caused the infection may impact on vaccine-induced neutralising responses (10.1126/science.abm0811). Do the authors know which variant caused the pre-infection in these individuals ? In addition, it was shown that the interval between the pre-infection and the vaccination could impact on the level of antibody responses (10.1016/S2666-5247(21)00275-5). How many weeks/months were the individuals infected before vaccination ?

4) Did the authors observe an impact of neutralising antibodies on the replication of variants ?

Minor comments:

5) How did the authors count the plaques ? Manually ? Blinded ? Were there several assessors ?

6) Supplementary data S1-S10: There are no error bars on the graphs. Do the authors show a representative assay for each individual ? Did the authors determine the neutralising titers for each individual in several independent experiments as well ?

7) Why did the authors decide to determine PRNT90 instead of PRNT50 ? Do not they have more chance to be on the plateau of the curve ?

8) Line 324: typo: "of" should be "or".

9) For the individuals who experienced a breakthrough infection, how many days post-2nd dose did they experience a breakthrough infection with Delta or Omicron ?

Reviewer #2:

Remarks to the Author:

The authors present a detailed investigation into the neutralization potency of a very well-constructed cohort of sera/plasma samples. This cohort brings together representative patient sample groups from numerous significant stages of the current SARS-CoV-2 pandemic and tests neutralization against the wt virus and all major VOCs. Vaccinated samples with and without viral infection are also included. These data provide numerous interesting and insightful details about the relationships of immune status with respect to neutralization. The authors use these data to establish antigenic relationships amongst the VOCs. These and other observations are incredibly well presented in the figures, making them accessible to a broad swath of the scientific community. By presenting such a comprehensive analysis and more stringent comparator criteria (PRNT90) than other studies in the literature, the authors have more persuasively shown the impact of Omicron on convalescent versus vaccinee sera. They observe that while vaccinee samples show a greater fold reduction, these begin at a much higher level of neutralizing potency and are less likely to be escaped entirely.

These data are timely, well presented, and accurately discussed. I have only the minor grammatical concerns listed below:

- Line 65: Omicron has of the most mutations observed so far, with the majority of them located in the spike protein.
 - o Change to: Omicron has the most mutations observed so far, with the majority of them located in the spike protein.
- Line 90-91: Mutations in the RBD pose the greatest risk increased infectivity or immune escape
 - o Change to: Mutations in the RBD pose the greatest risk for increased infectivity or immune escape
- Line 324: infection of vaccination ...
 - o Change to: infection or vaccination ...
- Line 368
 - o Remove the word Contrary

Reviewer #3:

Remarks to the Author:

Results are presented here of a set of very neatly executed experiments attempting to define the antigenic relationships between SARS-CoV-2 variants on the basis of plaque reduction neutralisation assays using both convalescent and vaccine sera.

Numbers are woefully small, but the data confirm a significant loss of neutralisation against omicron and also demonstrate a greater loss of neutralisation against zeta than to B.1, alpha, beta, gamma and delta among convalescent sera obtained from individuals infected with these earlier variants. A

similar pattern is observed among the vaccinated, and those who received the vaccine after having been naturally infected. Given the extremely low numbers, I would hesitate to present the data on the breakthrough infections, at least not in conjunction with the other two categories.

It is not clear why the WPV is n.a. for the 6 vaccinated patients who were previously infected, although the titres indicate that they were of the same order as the naïve vaccines. I would favour showing Fig 2A & B with Fig 1A-E (with the same scale) and combining Fig 1F and 2E into a single heatmap. This will make it more obvious that there is only a very small difference in drop in titre against omicron due to vaccination of those already infected in contrast to titres against the other variants which are generally much higher than among those with only natural immunity.

Performing antigenic cartography on such a small number of samples is a risky business but it does provide a nice way of visualizing the antigenic relationships and emphasizing that the progression has not been linear.

The paper contains several misunderstandings concerning the epidemiological and evolutionary processes underpinning the observations.

Ln 72 "therefore prevention and protection through vaccine-mediated immunity is still the preferred method for managing the pandemic"

– if "vaccine-mediated immunity" refers to infection, then this statement is incorrect.

Ln 435 "From an epidemiological point of view, this highlights that vaccine-induced immunity probably would have allowed lower virus circulation compared to immunity induced through infections, which may have prevented the evolution of new variants that harbor immune escape properties such as Omicron"

– vaccine-induced immunity (which is of shorter duration) is actually more likely to promote virus circulation compared to naturally acquired immunity, and the idea that the emergence of new variants can be prevented by keeping infection levels low is simplistic in the extreme.

Ln 461 "the continuous emergence of SARS-CoV-2 variants since late 2020 shows that the virus is still under significant evolutionary pressure and currently available vaccines might not be sufficient to mitigate the pandemic in the near-term future"

– SARS-CoV-2 will always be under significant evolutionary pressure but what remains to be seen is whether a single variant becomes dominant or whether there is continuous antigenic change.

– These vaccines do not durably prevent transmission so they cannot be used to mitigate the pandemic; they can be used to prevent severe disease and death but the data in this paper do not address the issue of whether that is variant-specific.

Ln 466 "Reassuringly, a combination of vaccine/infection derived immunity leads to broader antibody responses"

– It's not clear to me that this paper shows that.

REVIEWERS' COMMENTS

Reviewer #1 (Remarks to the Author):

"Bekliz et al. determined the ability of sera/plasma from convalescents infected with ancestral SARS-CoV-2, Alpha, Beta, Gamma or Delta to neutralise a panel of variants of concerns using plaque reduction neutralisation assay. Similar analysis was performed with sera/plasma from individuals vaccinated with 2 doses of mRNA vaccine with or without pre-infection and from vaccinees after a breakthrough infection with Delta or Omicron. Highest neutralization titers were observed against the homologous variant. The greatest immune escape was detected with Omicron. This immune escape was reduced in individuals who experienced a pre-infection before a double vaccination. Breakthrough infection with Omicron induced antibodies also able to neutralise Delta. The authors also performed an antigenic cartography of variants which identified Zeta and Omicron as separate antigenic clusters. This paper is very well written, clear and transparent. This paper is not the first one which compares heterologous humoral immunity but to my knowledge the other ones either compared less combinations or they used pseudoviruses (e.g. 10.1101/2022.01.28.477987, 10.1101/2022.01.03.21268582, 10.1126/science.abm0811). The plaque reduction neutralization assay is considered as the gold standard assay to determine neutralising responses. Data look accurate to me and robust. In addition, I had not seen any neutralisation data using Zeta isolate before this paper. The limitations are also well mentioned including the fact it is not a longitudinal study. Some preprints had shown that antibodies induced by an infection with Beta could neutralise better Gamma than Beta but they used pseudovirus neutralization assays (10.1101/2022.01.28.477987, 10.1101/2022.01.03.21268582). The results observed in this paper make more sense to me. Please find below some comments to potentially strengthen the paper. "

→ We thank the reviewer for the positive evaluation of our work and are grateful for the constructive criticisms. Please see below for detailed responses.

Response to the reviewer1:

Major comments:

1) "Do the authors have any RBD-specific IgG binding data ? Did the authors analyse the relationship between VOC RBD binding IgG and neutralization responses ? A paper which analysed vaccine-induced neutralising responses against VOCs in individuals with or without pre-infection with Alpha showed that S1 serology data using the Wuhan Hu-1 S1 RBD and VOC sequence was an unreliable marker for neutralization potency against variants of concerns (10.1126/science.abm0811). I think it would be valuable to compare the correlations between IgG binding data and neutralising responses in the context of the different combinations analysed in this paper."

→ *Response:* We thank the reviewer for this notion which is an important addition to our work. We fully agree that this data is an important aspect in this study.

For this purpose, all sera/plasma samples used in this work were analysed by the Meso Scale Discovery (MSD) COVID-19 serology kit that detects and quantifies anti-SARS-CoV-2 IgG specific for RBD-protein, with the exception of 8 sera out of a total of 120 for which no more sufficient volume was available. The Zeta variant was not available in this assay, hence we evaluated anti-IgG binding antibodies present in each sera/plasma to the RBD against ancestral SARS-CoV-2 (pre-VOC) and all VOCs (Alpha, Beta, Gamma, Delta and Omicron) used for PRNTs.

We have included this information in the materials and methods section (line 194-199), results section (line 332-375), discussion section (line 492-510) and in the supplementary data (Fig. S11-S12). We could confirm that correlation between RBD binding antibodies and neutralization for heterologous VOC was lower than for homologous viruses in most instances, but also observed a high variability between correlations for the different VOCs.

Of note, since the number of overall specimens used for post-Alpha, Beta, Gamma and post-vaccination with prior infection were already low and for some groups not enough volume was left

for further analysis, we have had only 7, 8, 8 and 6 for such convalescent sera, respectively. As a consequence, the low number of specimens clearly limits the validity of the correlation analysis and we have therefore excluded from the manuscript the results of the correlation of Alpha and post-vaccination with prior infection sera/plasma because they were quite homogeneous and could bias the results.

2) “Do the authors have similar data with mucosal samples (saliva, nasal samples) ? The comparison of systemic and mucosal neutralising responses would be very informative.”

→ *Response:* We agree with the reviewer that comparing the mucosal to systemic immune response would be informative, however unfortunately no mucosal samples were collected within the frame of this study. Nevertheless, we believe that our study is of high public health relevance, and adds important aspects to the understanding of the complex topic of pre-existing immunity. We acknowledge that it would be very interesting to do such a study in one of our future projects if we have the possibility to do so.

3) “The authors analysed the neutralising capacity of sera from vaccinees who were pre-infected before vaccination. The type of variant which caused the infection may impact on vaccine-induced neutralising responses (10.1126/science.abm0811). Do the authors know which variant caused the pre-infection in these individuals ? In addition, it was shown that the interval between the pre-infection and the vaccination could impact on the level of antibody responses (10.1016/S2666-5247(21)00275-5). How many weeks/months were the individuals infected before vaccination ?”

→ *Response:* Unfortunately, we only have this information for 3 participants that were infected during the 1st wave of COVID19 (March, October and December 2020). The other participants who had a subsequent infection with SARS-CoV-2 before vaccination were asymptomatic not aware of an infection at the time of serum collection. They were only identified as previously infected through the detection of anti-nucleocapsid antibodies in the frame of this study. Nevertheless, since their plasmas were collected in June 2021, we assume that these individuals were most likely infected with either Alpha or pre-VOC SARS-CoV-2 since no other variants have been widely circulating in Switzerland in that time period. For those individuals where prior infection was known, we have assessed this information and have added this information in the Table 1. The information is however not available for those that were only identified retrospectively by detection of nucleocapsid antibodies.

4) “Did the authors observe an impact of neutralising antibodies on the replication of variants ?”

→ *Response:* In this study we did not investigate the replication of the different SARS-CoV-2 variants neither in the presence nor in the absence of antibodies (sera/plasma). Since in the PRNT a reduced number of infectious foci is observed, it seems likely that neutralizing antibodies would have an impact on the replication of variants and reduce viral replication when present in the medium. However, in the assay we used in this study, the inoculum, which contains virus and serum/plasma dilutions, has been removed after the infection of cells and was replaced by an Avicel overlay that does not contain any neutralizing antibodies. Also, our only read-out was number of plaques after fixing the plates and we did not assess replication levels of virus in this assay. Nevertheless, we agree with the reviewer that this would be an interesting research question for a follow-up project.

Minor comments:

5) “How did the authors count the plaques ? Manually ? Blinded ? Were there several assessors ?”

→ *Response:* The plaques were all counted manually by a single assessor. However, when the plaques were not clear (e.g. tiny plaques or merging plaques...), experiments were repeated. In total, in this study we repeated 20 sera/plasma and the results were similar between replicates.

6) “Supplementary data S1-S10: There are no error bars on the graphs. Do the authors show a

representative assay for each individual ? Did the authors determine the neutralising titers for each individual in several independent experiments as well ?”

→ *Response*: For technical reasons we have unfortunately not replicated the experiments in several independent experiments. However, for each independent experiment, we used a standard reference serum to minimize inter-assay variation. In addition, as mentioned before, we have repeated 20 sera/plasma and we found similar results.

7) “Why did the authors decide to determine PRNT90 instead of PRNT50 ? Do not they have more chance to be on the plateau of the curve ?”

→ *Response*: In our study, we used PRNT90 to have a more stringent comparator criteria than other studies in the literature. This was also notified and highlighted as a strength of our study by reviewer# 2.

8) “Line 324: typo: “of” should be “or”.”

→ *Response*: Thank you, the text has been corrected.

9) “For the individuals who experienced a breakthrough infection, how many days post-2nd dose did they experience a breakthrough infection with Delta or Omicron ?”

→ *Response*: Of note, in this revised manuscript, we have increased the number of sera from Delta and Omicron breakthroughs individuals and the average of weeks between 2nd dose of vaccination and infection is 22 (range: 5-35) weeks for Delta breakthrough and 23 (range: 8-41) weeks for Omicron-breakthrough. Therefore, we have included this information in the updated Table 1. We hope that it is now clear.

Reviewer #2 (Remarks to the Author):

“The authors present a detailed investigation into the neutralization potency of a very well-constructed cohort of sera/plasma samples. This cohort brings together representative patient sample groups from numerous significant stages of the current SARS-CoV-2 pandemic and tests neutralization against the wt virus and all major VOCs. Vaccinated samples with and without viral infection are also included. These data provide numerous interesting and insightful details about the relationships of immune status with respect to neutralization. The authors use these data to establish antigenic relationships amongst the VOCs. These and other observations are incredibly well presented in the figures, making them accessible to a broad swath of the scientific community. By presenting such a comprehensive analysis and more stringent comparator criteria (PRNT90) than other studies in the literature, the authors have more persuasively shown the impact of Omicron on convalescent versus vaccinee sera. They observe that while vaccinee samples show a greater fold reduction, these begin at a much higher level of neutralizing potency and are less likely to be escaped entirely. These data are timely, well presented, and accurately discussed. I have only the minor grammatical concerns listed below:”

→ *Response*: We thank the reviewer for his words of appreciation and recognition of our work. We are grateful for his comments and feedback regarding our work.

- “Line 65: Omicron has of the most mutations observed so far, with the majority of them located in the spike protein.

- o Change to: Omicron has the most mutations observed so far, with the majority of them located in the spike protein.”

→ *Response*: Thank you, the text has been corrected in the revised manuscript.

- “Line 90-91: Mutations in the RBD pose the greatest risk increased infectivity or immune escape

- o Change to: Mutations in the RBD pose the greatest risk for increased infectivity or immune escape”

→ *Response: Response:* Thank you, the text has been corrected in the revised manuscript.

- “Line 324: infection of vaccination ...
o Change to: infection or vaccination ...”

→ *Response:* Thank you, the text has been corrected in the revised manuscript.

- “Line 368
o Remove the word Contrary”

→ *Response:* Thank you, the text has been corrected in the revised manuscript.

Reviewer #3 (Remarks to the Author):

“Results are presented here of a set of very neatly executed experiments attempting to define the antigenic relationships between SARS-CoV-2 variants on the basis of plaque reduction neutralisation assays using both convalescent and vaccine sera.

Numbers are woefully small, but the data confirm a significant loss of neutralisation against omicron and also demonstrate a greater loss of neutralisation against zeta than to B.1, alpha, beta, gamma and delta among convalescent sera obtained from individuals infected with these earlier variants. A similar pattern is observed among the vaccinated, and those who received the vaccine after having been naturally infected. Given the extremely low numbers, I would hesitate to present the data on the breakthrough infections, at least not in conjunction with the other two categories.”

→ *Response:* We thank the reviewer for this notion. We fully agree with the reviewer and we had increased the number of sera from Delta and Omicron breakthrough individuals to consolidate our findings.

“It is not clear why the WPV is n.a. for the 6 vaccinated patients who were previously infected, although the titres indicate that they were of the same order as the naïve vaccines.”

→ *Response:* We apologize for this oversight and have now completed the table (Table 1).

“I would favour showing Fig 2A & B with Fig 1A-E (with the same scale) and combining Fig 1F and 2E into a single heatmap. This will make it more obvious that there is only a very small difference in drop in titre against omicron due to vaccination of those already infected in contrast to titres against the other variants which are generally much higher than among those with only natural immunity.”

→ *Response:* We thank the reviewer for this suggestion. We have changed the figures accordingly. In order not to have overload the display items, we have combined all PRNT results in one figure and present the heatmap now as a separate figure. So now, we have figure 1 with the neutralization assays (PRNT90), figure 2 with the heatmap and figure 3 for the antigenic cartography.

“Performing antigenic cartography on such a small number of samples is a risky business but it does provide a nice way of visualizing the antigenic relationships and emphasizing that the progression has not been linear.”

→ *Response:* We thank the reviewer for his appreciation of the antigenic cartography. When comparing our approach to what has been published on the topic on SARS-CoV-2, we are in a comparable range for the number of human sera used by others. Indeed, Wilks, Samuel H *et al.*, *al* used 126 sera in total ¹ and Van der Straten *et al.*, used 51 convalescent and 112 post-vaccine specimens ². Of note, both other pre-prints used pseudoviral systems for assessment and not authentic virus isolates like we did.

Furthermore, we would like to highlight the logistical challenges to collect such convalescent sera from infected individuals with a proven SARS-CoV-2 infection for which the infecting strain was fully sequenced. This is especially true since some variants (Beta, Gamma) were not widely circulating in

Switzerland and such serum collections are highly dependent on the availability of patients (and their willingness to return for a blood draw for study purposes).

“The paper contains several misunderstandings concerning the epidemiological and evolutionary processes underpinning the observations.

Ln 72 “therefore prevention and protection through vaccine-mediated immunity is still the preferred method for managing the pandemic”

– if “vaccine-mediated immunity” refers to infection, then this statement is incorrect.”

→ *Response:* Vaccine mediated immunity refers to the immunity generated after vaccination and not infection. We apologize that we did not express ourself clearly and have now changed the wording which now reads « immunity generated through vaccination » in the updated manuscript.

“Ln 435 “From an epidemiological point of view, this highlights that vaccine-induced immunity probably would have allowed lower virus circulation compared to immunity induced through infections, which may have prevented the evolution of new variants that harbour immune escape properties such as Omicron”

– vaccine-induced immunity (which is of shorter duration) is actually more likely to promote virus circulation compared to naturally acquired immunity, and the idea that the emergence of new variants can be prevented by keeping infection levels low is simplistic in the extreme.”

→ We agree with the reviewer that we were too speculative and simplistic here and have therefore deleted the respective part: “High titers of neutralizing antibodies, such as elicited by the vaccine, were at least partially able to neutralize earlier variants before the rise of Omicron. From an epidemiological point of view, this highlights that vaccine-induced immunity probably would have allowed lower virus circulation compared to immunity induced through infections, which may have prevented the evolution of new variants that harbor immune escape properties such as Omicron.”

“Ln 461 “the continuous emergence of SARS-CoV-2 variants since late 2020 shows that the virus is still under significant evolutionary pressure and currently available vaccines might not be sufficient to mitigate the pandemic in the near-term future”

– SARS-CoV-2 will always be under significant evolutionary pressure but what remains to be seen is whether a single variant becomes dominant or whether there is continuous antigenic change.

– These vaccines do not durably prevent transmission so they cannot be used to mitigate the pandemic; they can be used to prevent severe disease and death but the data in this paper do not address the issue of whether that is variant-specific.”

→ *Response:* We agree with the reviewer and have deleted the sentence as it does not summarize the findings from our paper.

“Ln 466 “Reassuringly, a combination of vaccine/infection derived immunity leads to broader antibody responses”

– It’s not clear to me that this paper shows that.”

→ *Response:* In the antigenic map generated on specimen with hybrid immunity and vaccine breakthrough infections, of which we have added additional samples for the revision (panel B), we can see an orientation towards the middle of the map and towards other variants, which indeed indicates a broader neutralization. We therefore do believe that our statement is correct and hope that the reviewer agrees. This is also in line with what is stated by others that concluded that a hybrid immunity (infection + vaccination) can induce high quality antibodies with superior neutralization capacity against VOCs, including omicron ^{3,4}.

References

- 1 Wilks, S. H. *et al.* Mapping SARS-CoV-2 antigenic relationships and serological responses. *bioRxiv*, 2022.2001.2028.477987, doi:10.1101/2022.01.28.477987 (2022).
- 2 van der Straten, K. *et al.* Mapping the antigenic diversification of SARS-CoV-2. *medRxiv*, 2022.2001.2003.21268582, doi:10.1101/2022.01.03.21268582 (2022).
- 3 Carreno, J. M. *et al.* Activity of convalescent and vaccine serum against SARS-CoV-2 Omicron. *Nature* **602**, 682-688, doi:10.1038/s41586-022-04399-5 (2022).
- 4 Wratil, P. R. *et al.* Three exposures to the spike protein of SARS-CoV-2 by either infection or vaccination elicit superior neutralizing immunity to all variants of concern. *Nat Med* **28**, 496-503, doi:10.1038/s41591-022-01715-4 (2022).

Reviewers' Comments:

Reviewer #1:

Remarks to the Author:

The authors perfectly replied to all reviewers' comments.

They also significantly improved the manuscript by reorganizing the figures. They also added the analysis of "Correlation of receptor-binding-domain (RBD) binding antibodies with neutralizing titers" (S11) as I had suggested.

I only have one comment for the authors:

Line 198: could the authors mention the panel of the V-PLEX SARS-CoV-2 assay ? Is it panel 23 ? Or another one ?

Reviewer #3:

Remarks to the Author:

I am very happy with the corrections and believe the manuscript should be published without delay.

The authors perfectly replied to all reviewers' comments.

They also significantly improved the manuscript by reorganizing the figures. They also added the analysis of "Correlation of receptor-binding-domain (RBD) binding antibodies with neutralizing titers" (S11) as I had suggested.

➔ **Response:** We thank the reviewer for his supportive comment.

I only have one comment for the authors:

Line 198: could the authors mention the panel of the V-PLEX SARS-CoV-2 assay ? Is it panel 23 ? Or another one ?

➔ **Response:** We used the V-PLEX SARS-CoV-2 assay panel 22. We have added this information in the Methods section (line 534).